# Quest for the Co-Pyrolysis Behavior of Rice Husk and Cresol Distillation Residue: Interaction, Gas Evolution and Kinetics

**Na Xu [1], Mifen Cui [1], Zhuxiu Zhang [1,\*], Jihai Tang [1,2,\*] and Xu Qiao [1,2]**

[1] State Key Laboratory of Materials-Oriented Chemical Engineering, College of Chemical Engineering, Nanjing Tech University, Nanjing 211816, China; 201961104130@njtech.edu.cn (N.X.); mfcui@njtech.edu.cn (M.C.); qct@njtech.edu.cn (X.Q.)

[2] Jiangsu National Synergetic Innovation Centre for Advanced Materials (SICAM), No. 5 Xinmofan Road, Nanjing 210009, China

\* Correspondence: zhuxiu.zhang@njtech.edu.cn (Z.Z.); jhtang@njtech.edu.cn (J.T.)

**Abstract:** With the tremendous prosperity of industry, more and more hazardous waste is discharged from industrial production processes. Cresol distillation residue is a typical industrial hazardous waste that causes severe pollution without proper treatment. Herein, the co-pyrolysis of rice husk and cresol distillation residue was studied using thermogravimetry–mass spectrometry and kinetic studies. The Coats and Redfern method was employed to calculate the activation energy. The results indicated that the pyrolysis process of cresol distillation residue and RH/CDR (Rice Husk and Cresol Distillation Residue) blends can be divided into four stages and three stages for RH. The introduction of RH not only improved the thermo-stability of cresol distillation residue at a low temperature but also reduced the activation energy of the blends. The activation energy was the lowest when the proportion of rice husk in the blend was 60%. The main gaseous pyrolysis products included $CH_4$, $H_2O$, $C_2H_2$, $CO_2$, $C_3H_6$ and $H_2$. There existed an unusual combination of synergistic and inhibitive interactions between RH and cresol distillation residue, respectively, within different temperature ranges. The synergistic interaction decreased the reaction's activation energy, whereas the inhibitive interaction reduced the emission of main gaseous products, such as $CH_4$ and $CO_2$. It was concluded that the addition of RH was conducive to improving the pyrolytic performance of cresol distillation residue and the resource utilization of cresol distillation residue.

**Keywords:** co-pyrolysis; cresol distillation residue; rice husk; TG-MS; kinetics

## 1. Introduction

Distillation processes dominate 60% of separation in the chemical industry [1], but there are ca. 2.5 million tons of distillation residues produced in China every year. Distillation residues have been included in the national hazardous waste lists of different countries [2]. In general, distillation residue is mainly treated by landfill or incineration approaches [3–5]. The landfill of distillation residues generates large amounts of leachate that severely contaminate the soil and underground water [6–9]. The incineration of distillation residues is a high-energy consumption process that also produces severe secondary pollution [10,11]. The resource utilization of distillation residues for the production of value-added products seems to be an environment-benign approach [12], but only a handful of papers exist regarding the conversion of distillation residues to diesel and lubricating oil [13].

At present, varieties of biomass with clean and renewable characteristics have been used for the resource utilization of solid waste, including sewage sludge, food waste and municipal solid waste, through co-pyrolysis technology [14–17]. There is an inhibitive or synergistic interaction between biomass and solid waste during thermal treatment. We note that only one paper reported an inhibitive interaction between solid waste and biomass, in which the undecomposed lignite particles prevented the release of volatile matters in

solid waste derived from refining and chemical wastewater at a lower temperature [18]. In contrast, synergistic interactions have been extensively studied. Several researchers have reported a synergistic interaction between sewage sludge and biomass that reduced the release of gaseous sulfur substances and NOx [19–21]. The structure of pyrolysis products can be optimized through synergistic interactions, as exemplified by the improved surface area of combustion ashes from textile dyeing sludge [22]. In addition, the synergistic interactions of blends have also resulted in higher reactivity and better combustion performance. For example, the synergistic interactions between textile dyeing sludge and microalgae improve the combustion performance of textile dyeing sludge because the density of its blends is larger than single microalgae [23]. We note that the co-pyrolysis of industrial distillation residues with biomass remains relatively underexplored. There are fewer than five studies focusing on co-pyrolysis of biomass with the distillation residue from lab-scale bio-oil production [24–27]. The interaction between industrial distillation residues and biomass remains relatively unexplored.

In this contribution, we reported an unexpected synergistic effect that combined the high-temperature inhibitive and low-temperature synergistic processes in a sequential manner during the co-pyrolysis of cresol distillation residue and rice husk. Cresol distillation residue is a typical industrial waste from the production of p-cresol, which has been widely used for the synthesis of pharmaceuticals, herbicides, antioxidants and dyes [28]. It is estimated that 10–12 tons of cresol residue can be produced for every 100 tons of p-cresol [29]. This work aims to investigate the interactions and product characteristics during co-pyrolysis of cresol distillation residue (CDR) and rice husk (RH) at various mixing ratios through thermogravimetric analysis. The interactions between cresol distillation residue and rice husk were investigated using the deviation of weight loss TG ($\Delta W$) between the calculated and experimental values in detail. TG coupled with mass spectrometry (TG-MS) enables the tracing of thermal reactions and the characterization of evolved gases. The kinetic parameters and apparent activation energy during thermal decomposition were calculated using the Coats and Redfern model. The elucidation of interactions between RH and CDR during the co-pyrolysis process is likely to provide scientific support for the effective utilization of CDR and to reduce related environmental hazards.

## 2. Experimental

### 2.1. Materials

The cresol distillation residue was supplied by a local company in Nantong, and rice husks were purchased from an agricultural product store. RH was mixed with CDR in a ratio of 20–80% to examine the effect of biomass additives on thermal decomposition during the co-pyrolysis process. Residue–biomass mixtures were prepared with a biomass content of 20%, 40%, 60% and 80%. All samples were oven-dried at 105 ± 5 °C for 24 h to remove water before pyrolysis.

The elemental content and chemical features of raw materials are exhibited in Table 1. The ash content, volatile matter and moisture were measured following the Chinese coal industry method (Chinese standard methods, GB/T 212-2008). The elemental analysis was performed using an elemental analyzer (Euro Vector EA3000, NETZSCH, Italy). The element ratio of H/C was adopted to characterize the amount of $CO_2$ released during pyrolysis [30]: the higher the H/C, the lower the level of $CO_2$ per unit of energy produced [31]. In this work, it was demonstrated that in comparison to cresol distillation residue, rice husk has potentially lower $CO_2$ emissions. The elemental content of Ca, Fe, Al, K, P, Cl and Si were determined using X-ray fluorescence (Table 2).

**Table 1.** Proximate analysis data for raw materials.

| Samples | Ultimate Analysis (%) | | | | | Proximate Analysis (%) | | | | H/C |
|---|---|---|---|---|---|---|---|---|---|---|
| | C | N | H | S | O | A | VM | FC | M | |
| RH | 42.68 | 0.54 | 5.26 | 0.06 | 37.78 | 13.68 | 70.45 | 9.45 | 6.42 | 0.1232 |
| CDR | 64.12 | 0 | 6.23 | 3.57 | 25.53 | 0.55 | 60.00 | 29.45 | 10.00 | 0.0972 |

Calculated by O = 100-Ash-C-H-N-S. A = ash; VM = volatile matter; FC = fixed carbon; M = moisture.

**Table 2.** XRF analysis results of cresol distillation residue and rice husk.

| Samples | Content(%) | | | | | | | |
|---|---|---|---|---|---|---|---|---|
| | Ca | Mg | K | Fe | Cu | Zn | Ni | Si |
| RH | 0.1222 | 0.0708 | 0.6647 | 0.0279 | 0.0014 | 0.0038 | 0.0022 | 6.0389 |
| CDR | 0.0025 | 0.0215 | 0.0021 | 0.0012 | | | | 0.0814 |

*2.2. Methods*

2.2.1. TG-MS

CDR was mixed with RH ground into a particle size of less than 100 mesh via vigorous stirring, with an RH weight percentage of 20–80 wt.%. Pyrolysis of all samples was performed on STA 449-QMS 403 TG-MS analyzers. About $10 \pm 0.5$ mg samples with particle sizes less than 100 mesh were heated from room temperature to 1000 °C with a heating rate of 20 °C/min and a $N_2$ flow rate of 20 mL/min. The generated gaseous products were monitored, including $H_2$ ($m/z = 2$), $CH_4$ ($m/z = 16$), $H_2O$ ($m/z = 18$), $C_2H_2$ ($m/z = 26$), $C_3H_6$ ($m/z = 42$) and $CO_2$ ($m/z = 44$), according to the database of the National Institute of Standards and Technology (NIST). All the pyrolysis test conditions were repeated two or three times, and the average data were taken to ensure repeatability.

2.2.2. Kinetic Analysis

We selected the Coats and Redfern model to calculate kinetic parameters based on TG analysis [32,33].

The conversion fraction is the mass fraction of a decomposed sample [34,35], which is described by:

$$x = \frac{w_0 - w_t}{w_0 - w_\infty} \tag{1}$$

where $x$ is the conversion extent of samples, $W_0$ = initial mass, $W_t$ = instantaneous mass and $W_\infty$ = final mass.

Isothermal reaction rate:

$$\frac{dx}{dt} = A_0 e^{-\left(\frac{E}{RT}\right)} (1-x)^n \tag{2}$$

A constant heating rate:

$$\beta = \frac{dT}{dt} \tag{3}$$

Thus, Equation (2) is also equal to the following equation:

$$\frac{dx}{dT} = \frac{A_0}{\beta} e^{-\left(\frac{E}{RT}\right)} (1-x)^n \tag{4}$$

The integral of Equation (4):

$$\frac{1 - (1-x)^{1-n}}{1-n} = \frac{A_0}{\beta} \int_0^T e^{-\left(\frac{E}{RT}\right)} dT \tag{5}$$

First, perform partial integration on the right side of Equation (5); then, ignore the higher-order terms to obtain the following expression:

$$\frac{1-(1-x)^{1-n}}{1-n} = \frac{A_0 R T^2}{\beta E}\left[1-\left(\frac{2RT}{E}\right)\right]e^{(-E/RT)} \quad n \neq 1 \tag{6}$$

$$-\ln(1-x) = \frac{A_0 R T^2}{\beta E}\left[1-\left(\frac{2RT}{E}\right)\right]e^{(-E/RT)} \quad n = 1 \tag{7}$$

Equations (6) and (7) can be written as a logarithm:

$$\ln\left[\frac{1-(1-x)^{1-n}}{T^2(1-n)}\right] = \ln\left[\frac{A_0 R}{\beta E}\left(1-\left(\frac{2RT}{E}\right)\right)\right] - \frac{E}{RT} \quad n \neq 1 \tag{8}$$

$$\ln\left[\frac{-\ln(1-x)}{T^2}\right] = \ln\left[\frac{A_0 R}{\beta E}\left(1-\left(\frac{2RT}{E}\right)\right)\right] - \frac{E}{RT} \quad n = 1 \tag{9}$$

Because $2RT/E \ll 1$, we simplify Equations (8) and (9):

$$\ln\left[\frac{1-(1-x)^{1-n}}{T^2(1-n)}\right] = \ln\left[\frac{A_0 R}{\beta E}\right] - \frac{E}{RT} \quad n \neq 1 \tag{10}$$

$$\ln\left[\frac{-\ln(1-x)}{T^2}\right] = \ln\left[\frac{A_0 R}{\beta E}\right] - \frac{E}{RT} \quad n = 1 \tag{11}$$

It is assumed that the pyrolysis reactions follow the first-order reaction kinetics at $n = 1$, which gives:

$$\ln\left[\frac{-\ln(1-x)}{T^2}\right] = \ln\left[\frac{A_0 R}{\beta E}\right] - \frac{E}{RT} \tag{12}$$

For easy calculating, Equation (12) can be changed to the straight-line formula:

$$Y = \ln\left[-\frac{\ln(1-x)}{T^2}\right], X = \frac{1}{T} \tag{13}$$

$x$ = instantaneous conversion ratio, $T$ = absolute temperature (K), $\beta$ = heating rate (°C/min), $R$ = Gas constant (J·mol$^{-1}$·K$^{-1}$), $A_0$ = pre-exponential factor (min$^{-1}$) and $E$ = activation energy (K·J·mol$^{-1}$).

## 3. Results and Discussion

### 3.1. Thermogravimetric Analysis

3.1.1. Individual Samples

Thermogravimetric (TG) and differential thermal gravity (DTG) curves showed a difference in thermal behaviors between RH and CDR (Figure 1). RH consists of (hemi)cellulose, lignin and other organic minorities [36,37] so that the weight loss of RH can be divided into three main stages (Figure 1a). The release of adsorbed gases and water vapor occurred from 50 to 218 °C, followed by the thermal decomposition of volatiles from (hemi)cellulose with the range of 218 to 409 °C (weight loss = 57.3%). The final stage is the slow decomposition of lignin and carbonaceous residue with a weight loss of 14.9%. In comparison, the pyrolysis of CDR was divided into four individual stages (Figure 1b). Water evaporation occurred before 129 °C, after which the pyrolysis of (hemi)celluloses, phenols and ethers occurred until 510 °C. The weight loss from 510 °C to 750 °C corresponded to the charring reaction of lignin and the remaining hydrocarbon, followed by the thermal decomposition of inorganic substances at a very low rate [38]. As the DTG curves show, the maximum weight loss rate of RH (19.16 wt.%/min) is much higher than CDR (9. 4 wt.%/min) (Table 3), indicating the higher reactivity of RH than CDR during the co-pyrolysis process. Compared with RH, the peak temperature of maximum weight loss of CDR was lower than that of RH,

indicating that CDR was more unstable during the co-pyrolysis process. Therefore, RH exhibited better pyrolysis performance than CDR, and it is estimated that the blends of RH with CDR enhance the pyrolysis process of CDR.

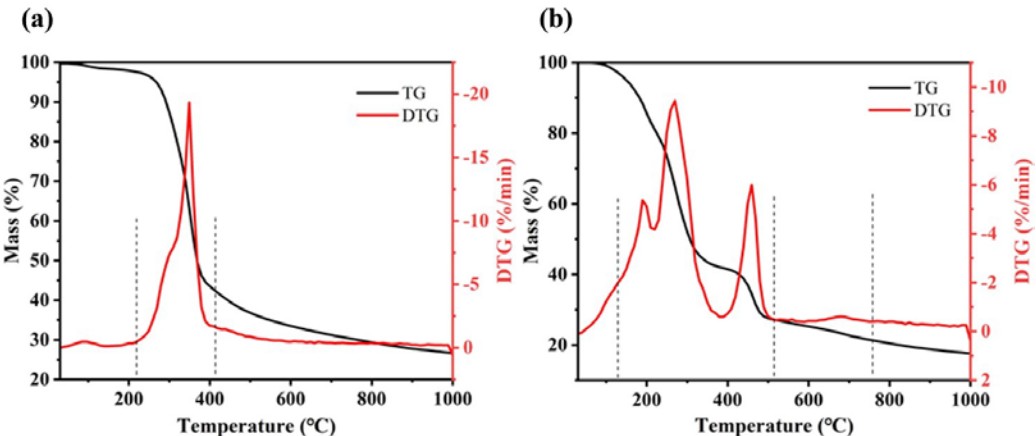

**Figure 1.** TG and DTG curves of RH (**a**) and CDR (**b**) from 25 to 1000 °C.

**Table 3.** Pyrolysis parameters for all samples.

| Samples | $T_i$ (°C) | $T_f$ (°C) | $T_{max}$ (°C) | $DTG_{max}$ (wt.%/min) | $M_f$ % |
|---|---|---|---|---|---|
| RH | 274.20 | 983 | 350 | 19.16 | 26.66 |
| 80 RH | 249.03 | 982 | 320 | 9.06 | 25.21 |
| 60 RH | 222.54 | 978 | 298 | 8.37 | 25.44 |
| 40 RH | 222.11 | 975 | 288 | 8.80 | 26.09 |
| 20 RH | 194.09 | 973 | 270 | 9.46 | 26.97 |
| CDR | 177.24 | 965 | 268 | 11.84 | 17.58 |

$T_i$ is the initial decomposition temperature; $T_f$ is the final decomposition temperature; $T_{max}$ is the first peak temperature; $DTG_{max}$ is the maximum weight loss rate; $M_f$ is the residue mass.

The uncertainty values $u_1$, $u_2$, $u_3$, $u_4$, $u_5$ and $u_6$ are introduced by the measurement error of RH, 80RH, 60RH, 40RH, 20RH and CDR [39]. $u$ is written as:

$$u = \frac{\delta}{k} \tag{14}$$

where $\delta$ is error, and $k = \sqrt{3}$.

We calculate the error with $M_f$ as the reference:

$$\delta = \left| M_f - \overline{M_f} \right|$$
$$\overline{M_f} = 24.66 \tag{15}$$

$$u_1 = 1.15, u_2 = 0.32, u_3 = 0.45, u_4 = 0.83, u_5 = 1.33, u_6 = 4.09$$

The larger the uncertainty value is, the more unstable the sample is. The most unstable sample is CDR, which is consistent with the above analysis.

### 3.1.2. Blend Samples

The TG/DTG curves of the blends of RH and CDR can be roughly divided into four stages, similar to CDR, but the beginning temperature ($T_i$), maximum decomposition rate ($T_{max}$) and ending temperature ($T_f$) of the blends are totally different (Figure 2a,b, Table 3). The addition of RH to CDR reduced the heat transfer from the surface to the core of the blend samples, giving rise to the shift of both blends' $T_i$ and $T_f$ to higher temperatures [40]. The addition of RH significantly reduced the maximum rate of weight loss ($DTG_{max}$) and residual mass ($M_f$) of the blend samples from 20RH to 60RH. We noted that the $DTG_{max}$ peak gradually moved toward the high-temperature zone with the increase in RH content,

indicating the addition of RH reduced the reactivity of the mixtures. However, when the proportion of RH is 80%, the DTG curve of 80RH/CDR is similar to pure RH. In Figure 2c, the conversion of all samples occurred from 100 to 900 °C. The conversion curve of the mixture almost coincides with the RH curve from 372 to 472 °C. The pyrolysis of all samples was delayed when the temperature rose above 472 °C, corresponding to the highly reduced value of the second DTG peak in Figure 2b.

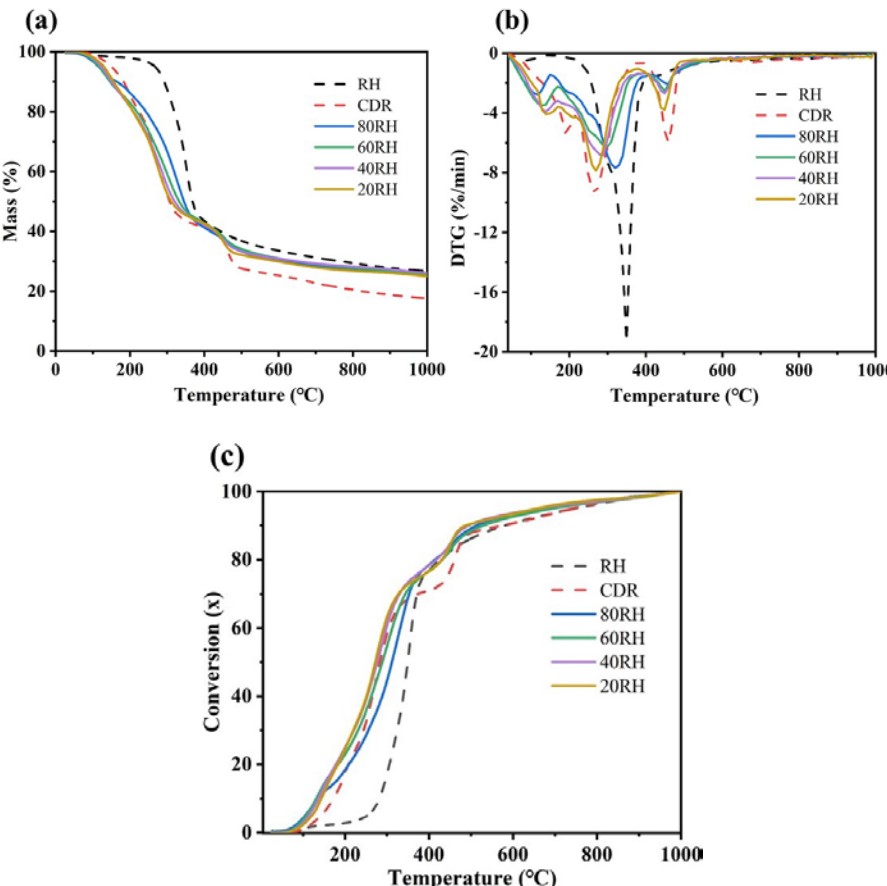

**Figure 2.** (**a**) TG curves; (**b**) DTG curves; and (**c**) conversion degrees of RH, CDR and blends at 20 °C/min heating rate.

### 3.2. Interactive Effects Analysis

For the purpose of investigating the interactions between RH and CDR during co-pyrolysis, the weight loss deviations between experimental and calculated values within the whole temperature range were calculated according to the following equation [41,42]:

$$w_{calculated} = x_1 \times w_{RH} + x_2 \times w_{CDR} \tag{16}$$

$$\Delta W = W_{experimental} - W_{calculated} \tag{17}$$

where $W_{RH}$ and $W_{CDR}$ represented the weight loss (TG) of each sample. $x_1$ and $x_2$ were the proportion of RH and CDR in the blend, respectively. $W_{calculated}$ is the sum of the component weight based upon its fraction at certain temperatures [43–45]. $\Delta W$ refers to the deviation of weight loss of the blend according to the TG curves, which could be used as an indicator of the interaction degree. There is no interaction if the value of $\Delta W$ is 0 [46].

Three stages of co-pyrolysis could be identified from Figure 3. The negative $\Delta W$ value reflects the synergistic effect between RH and CDR at temperatures below 374 °C due to the catalytic effect of alkali and alkaline earth metals in the rice husk (Table 2) [47,48]. In comparison, the calculated TG curves were below the experimental TG curves above 455 °C, demonstrating an inhibitive effect between RH and CDR during pyrolysis. The

inhibitive effect is ascribed to the adherence of CDR pyrolysis products to the blend's surface, which, in turn, prevented further volatilization and attenuated the heat/mass transfers [49]. Moreover, it is the first time a perfect match between the calculated TG curves and the experimental TG curves between 374 and 455 °C has been reported, indicating the complete degradation of all volatiles generated from the initial pyrolysis of blends in this temperature range.

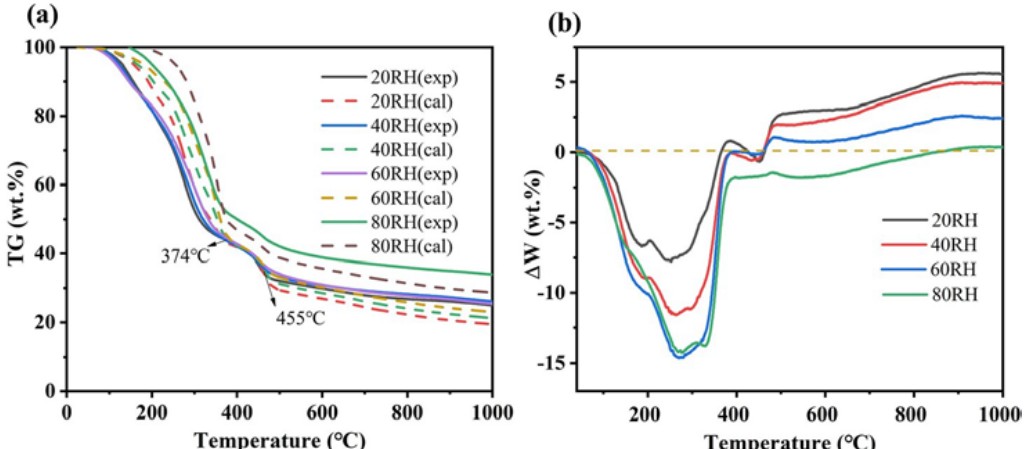

**Figure 3.** The comparison (**a**) and deviation (**b**) of experimental and calculated TG curves.

*3.3. TG-MS Analysis*

The ion fragments of the main gaseous products generated during pyrolysis were monitored by TG-MS analysis (Figure 4). $CH_4$, $H_2O$ and $CO_2$ were major products during the pyrolysis process. $CH_4$ has three ionic strength peaks in the range of 100–309 °C, 309–478 °C and 478–900 °C, respectively. The synergistic stage from 100 °C to 309 °C is associated with the cracking of side chains of aliphatic hydrocarbons. The conversion of long-chained aromatic groups and alkyl groups to methane then occurred from 309 °C to 478 °C [50]. In the inhibitive stage, $CH_4$ is mainly derived from the conversion of methoxyl groups in lignin after 478 °C [51,52]. The $CH_4$ peak intensity in the synergistic zone was lower than that in the inhibitory zone, which is mainly attributed to the speed of weight loss. However, the amount of $CH_4$ released in both zones decreased after adding rice husk.

$H_2O$ evolution can also be divided into three stages. $H_2O$ peaks in the synergistic region (<317 °C) are attributed to the loss of cellular water and the hydration of the phenolic hydroxyl groups from chemically bonded water and distillation residue [53]. Then, the thermal decomposition of light-volatile components occurred from 317 to 470 °C. The release of $H_2O$ in the inhibitive region (>470 °C) resulted from the binding of some free radical groups, such as hydroxide radicals, and oxygen ions, as well as the decomposition of O-contained functional groups (especially hydroxyl groups) [54]. During the whole pyrolysis process, the amount of $H_2O$ released from blends 60RH and 80RH was always higher, as opposed to 20RH and 40RH.

The concentration of $CO_2$ initially increased in the synergistic region (300–450 °C) because of the breaking of aromatic moieties and carboxyl groups. The synergistic effect led to the rapid release of $CO_2$. Then, the second peak of $CO_2$ was observed between 450 °C and 700 °C, which resulted from the degradation of carbonyl compounds and oxygenated compounds with high thermal stability [55]. When the temperature reached the inhibitive region (>700 °C), the decrease in $CO_2$ indicated the decomposition of a small amount of $CaCO_3$ [56]. However, the $CO_2$ emission of blends decreased compared with CDR. The $CO_2$ emission was the lowest at 60RH due to the inhibitive interaction.

The $H_2$ emission occurred continuously over the temperature range of 300–900 °C (single peak) for each sample. The degradation of hydrogen-rich compounds occurred at about 300 °C, and the condensation of (hydro)aromatic compounds or the decomposition of heterocyclic compounds were detected above 600 °C [53,57]. The hydrogen release of

CDR is higher than RH owing to the higher hydrogen content in CDR (Table 1). For blends, the peak values of $H_2$ occurred in the order of 40RH, 20RH, 0RH, 60RH, 80RH and 100RH, which differed from the regular sequence of the RH-addition ratio. Additionally, 20RH and 40RH contributed to the promotion of $H_2$ production, while other mixtures were just the opposite. $H_2$ yield was highest when the RH-addition ratio was 40 wt% (40RH).

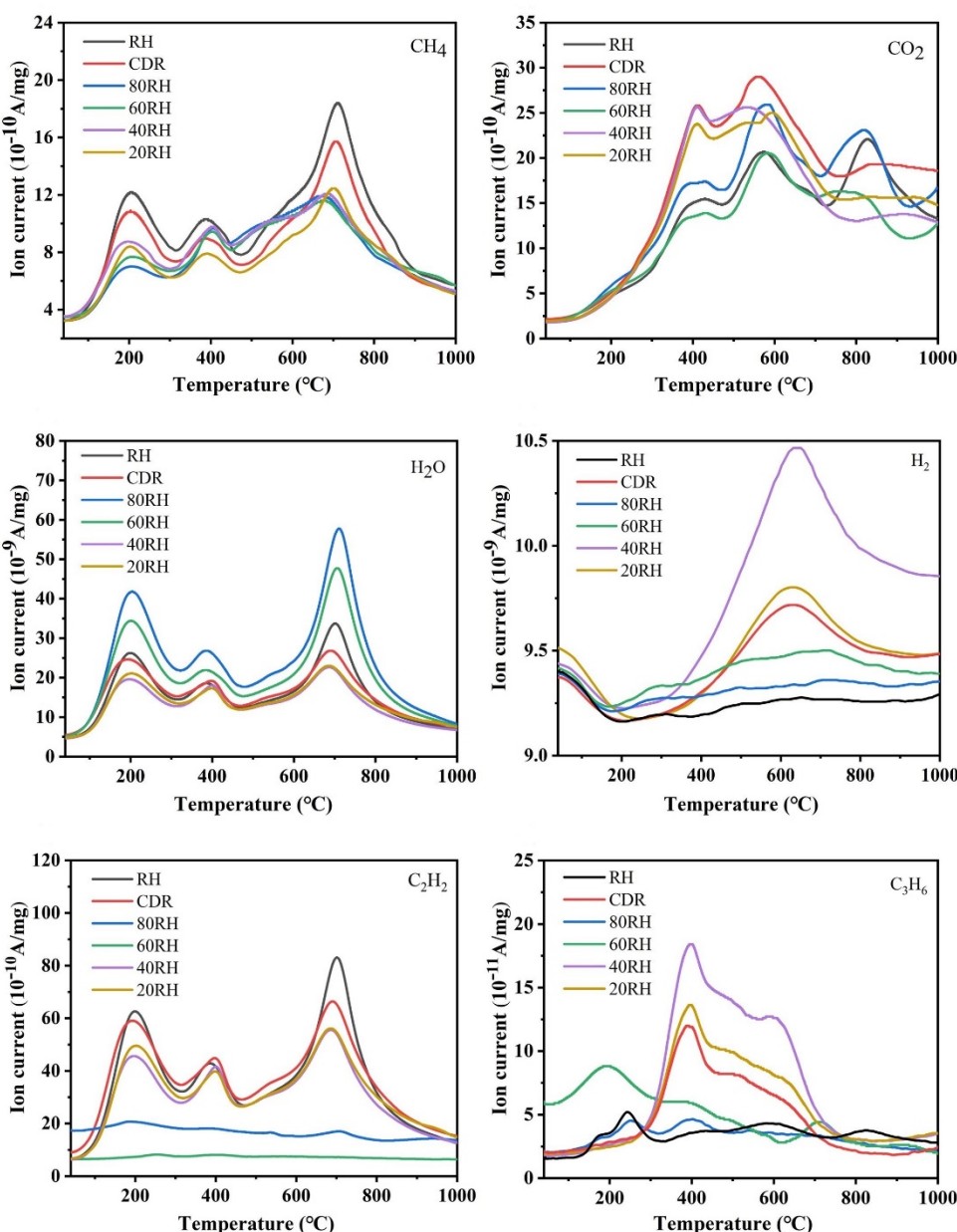

**Figure 4.** Gas emission curves during pyrolysis process.

Aliphatic hydrocarbons $C_nH_m$ ($n \geq 2$) can be generated by two pathways. One is the decomposition of macromolecular components, including long-chain and branched-chains paraffin in CDR, into small molecules. Another is the degradation of hydroaromatic groups, polymethylene and aliphatic bridges (e.g., *n*-fatty acis). Alkyne ($C_2H_2$) and alkene ($C_3H_6$) have strong ionic peaks within the temperature range of 200–700 °C.

### 3.4. Kinetic Analysis

All kinetic parameters were calculated via the Coats and Redfern method within the temperature ranges from 190 to 380 °C and from 260 to 400 °C for CDR and RH, respectively. According to Equation (12), a series of linear-fitting curves were obtained and plotted in

Figure 5. The activation energy $E_a$ and the pre-exponential factor $A_0$ were thereby generated for various blend ratios (Table 4). The activation energies for RH, CDR and their blends lie in the range of 15–25 kJ/mol. The linear correlation coefficient $R^2$ of 0.95 indicates the reasonable fitting of the first-order reaction model. As exhibited in Table 4, the $E_a$ values of RH and CDR were 21.85 kJ/mol and 24.00 kJ/mol, respectively. The $E_a$ values of blends with different proportions were lower than those of RH or CDR, demonstrating that the existence of RH could reduce the energy required for CDR pyrolysis. This result also verified that a higher value of activation energy indicates a slower reaction. As the activation energy represents the critical energy required to initiate a reaction [58], the 60 RH blend, with the lowest activation energy among these blended samples, is recommended. It is consistent with the positive synergistic effect induced by the addition of RH so that the optimal mixing ratio of RH to CDR is 3:2.

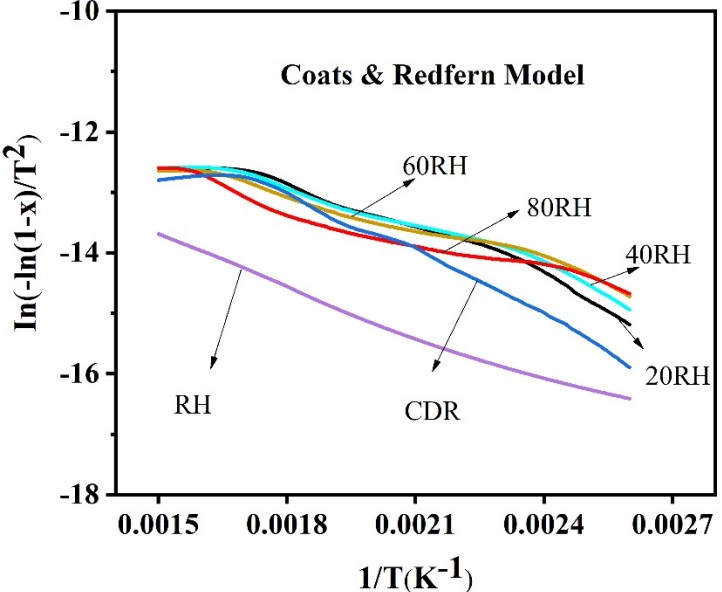

**Figure 5.** Plot of $\ln\left[-\ln(1-x)/T^2\right]$ vs. $1/T$ for the pyrolysis process of RH/CDR blends.

**Table 4.** Kinetic parameters for the pyrolysis of RH, CDR and their blends.

| Samples | Equation | $E_a$ (kJ/mol) | $A_0$ (min$^{-1}$) | $R^2$ |
|---|---|---|---|---|
| RH | $y = -2628.6x - 9.8112$ | 21.85 | 42.25 | 0.9918 |
| 80 RH | $y = -1852.8x - 9.9077$ | 15.40 | 27.04 | 0.9530 |
| 60 RH | $y = -1805.5x - 9.8152$ | 15.01 | 28.91 | 0.9845 |
| 40 RH | $y = -2015.3x - 9.3487$ | 16.76 | 51.46 | 0.9774 |
| 20 RH | $y = -2223x - 8.9758$ | 18.48 | 82.38 | 0.9597 |
| CDR | $y = -2887.8x - 8.0009$ | 24.00 | 283.62 | 0.9649 |

## 4. Discussions

This work will be discussed from the following two points: (1) interaction, (2) Pyrolysis products, and (3) kinetics data.

(a) Interaction: The deviation of weight loss TG ($\Delta W$) demonstrated that there were synergistic interaction, no interaction and inhibitive interaction between RH and CDR at 76–374 °C, 374–455 °C and 455–1000 °C, respectively. Low temperatures favor synergistic interaction, which is consistent with some previous studies [26,59]. It is reported that the synergistic mechanism was mainly attributed to the catalytic effects of alkali and alkaline earth metals and the transfer of hydrogen and hydroxy radicals [60]. The phenomenon of no interaction generally occurs at the initial stage of pyrolysis due to the low temperature, at which the sample has not started to degrade yet [61]. It is a new discovery that there is no interaction in the middle temperature.

This may be the reason for the temporary pause of the blends as the volatiles decrease. The inhibition mechanism was mainly attributed to the carbonization of biomass at high temperatures [62]. Further decomposition of CDR was hindered by a large number of carbonaceous deposits that covered and blocked the molecule pores of CDR residues.

(b)  Pyrolysis products: All co-pyrolysis products including $CH_4$, $H_2O$, $CO_2$, $H_2$ and light hydrocarbon were detected via MS. The addition of rice husk reduced the main gaseous products $CH_4$ and $CO_2$. For $CH_4$, RH consistently produced more methane than CDR. This result was mainly attributed to the removal of methoxyl substituents of the lignin, cellulose and hemicellulose and the conversion of the alkyl chain of the lignin [63]. CDR was dominant during the pyrolysis of the blends, which reduced methane production. CDR produces a large amount of $CO_2$ between 400 °C and 600 °C, indicating that a large number of aliphatic groups in CDR were produced by decarboxylation/decarbonylation reaction [64].

(c)  Kinetics data: The activation energy of RH in non-catalytic pyrolysis was 21.85 kJ/mol, which is far lower than the results of other studies. Balasundram et al. [65] revealed that the activation energy of RH under non-catalytic action was 49.78 kJ/mol, lower than 53.10 kJ/mol under catalytic action. The kinetic study of CDR has not been reported before this work. López-González et al. [66] reported activation energy of some biomass samples, such as Nannochloropsis gaditana, Scenedesmus almeriensis and Chlorella vulgaris, during pyrolysis in the range of 135–178 kJ/mol. Zhu et al. [26] reported an activation energy value of 71 kJ/mol for bio-oil distillation residue. Sanchez et al. [67] reported that the activation energy of animal manure, sewage sludge and municipal solid waste are 140, 143 and 173 kJ/mol. All samples studied in this paper have low activation energies, mainly due to the synergistic interaction at low temperatures. The synergistic interaction promoted the reaction process and resulted in a significant decrease in activation energy in corresponding conversion stages [62].

Therefore, the future research direction of distillation residues or other industrial hazardous waste can be started from the perspective of interaction and pyrolysis products. In short, compared with the traditional treatment methods, such as landfilling or incinerating, the co-pyrolysis of industrial hazardous waste and biomass would be a better solution.

## 5. Conclusions

In summary, an unexpected interaction existed in the co-pyrolysis of CDR and RH. There is a synergistic interaction between RH and CDR from 76 to 374 °C, which disappears in the medium temperature range. The inhibitive interaction occurs from 500 to 1000 °C. All co-pyrolysis products, including $CH_4$, $H_2O$, $CO_2$, $H_2$ and light hydrocarbon, were detected via MS. Inhibitive interactions reduced the main gaseous product ($CH_4$ and $CO_2$), and synergistic interactions decreased the activation energy simultaneously. The optimum blending ratio between RH and CDR based on the lowest activation energy of 15.01 kJ/mol is 3:2. The interaction, gas evolution and kinetic parameters will be helpful to large-scale co-pyrolysis of cresol distillation residue and rice husk, and they also provide a promising solution for other distillation residues.

**Author Contributions:** N.X.: Methodology, Validation, Investigation, Writing—original draft, Software, Formal analysis. Z.Z.: Writing—review and editing, Supervision, Data curation, Funding acquisition. J.T.: Writing—review and editing, Funding acquisition. M.C.: Writing—review and editing. X.Q.: Supervision, Writing—review and editing, Funding acquisition. All authors have read and agreed to the published version of the manuscript.

**Funding:** We gratefully acknowledge the financial support from National Key R&D Program of China (2021YFE0191100), National Natural Science Foundation of China (21808104, 21676141), and Key R&D Program of Jiangsu Province (BE2021710).

**Acknowledgments:** The authors gratefully acknowledge the financial support from the National Key R&D Program of China (2021YFE0191100), Provincial Key R&D Program of Jiangsu (BE2021710) and South Africa/China Joint Research Programme (CHIN200227507231).

**Conflicts of Interest:** The authors declare no conflict of interest.

**Nomenclatures**

| | |
|---|---|
| CDR | Cresol distillation residue |
| RH | Rice husk |
| TG-MS | thermogravimetry–mass spectrometry |
| $\Delta W$ | the deviation of weight loss TG |
| NIST | National Institute of Standards and Technology |
| TG | Thermogravimetric |
| DTG | Differential thermal gravity |
| A | Ash |
| VM | Volatile matter |
| FC | Fixed carbon |
| M | Moisture |
| $T_i$ | the initial decomposition temperature (°C) |
| $T_f$ | the final decomposition temperature (°C) |
| $T_{max}$ | the peak temperature (°C) |
| $DTG_{max}$ | the maximum weight loss rate (wt.%/min) |
| $M_f$ | the residue mass (%) |
| CPI | comprehensive pyrolysis index |
| $x$ | the rate of conversion |
| $\beta$ | heating rate |
| $E_a$ | activation energy (kJ/mol) |
| $A_0$ | pre-exponential factor (min$^{-1}$) |
| $R^2$ | linear correlation coefficient |

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
