# Peer review of "Quest for the Co-Pyrolysis Behavior of Rice Husk and Cresol Distillation Residue: Interaction, Gas Evolution and Kinetics"

_energies, doi:10.3390/en15062130_

Round 1

Reviewer 1 Report

In general, the topic of this paper is interesting. The subject could be relevant and appropriate for the Energies. Here follow some points that need further attention:

-  The abstract is too short. A single paragraph of 200 words is suggested. The abstract should have the following structure: (1) Background: Place the question addressed in a broad context and highlight the purpose of the study; (2) Methods: briefly describe the main methods or treatments applied; (3) Results: summarize the article’s main findings; (4) Conclusions: indicate the main conclusions or interpretations.

- The authors should cite more papers related to rice husk and cresol distillation residue. Here are some suggestions:

  1. Gao, W., Li, H., Song, B., & Zhang, S. (2020). Integrated leaching and thermochemical technologies for producing high-value products from rice husk: leaching of rice husk with the aqueous phases of bioliquids. Energies13(22), 6033.
  2. Hu, X., Zhang, Z., Gholizadeh, M., Zhang, S., Lam, C. H., Xiong, Z., & Wang, Y. (2020). Coke formation during thermal treatment of bio-oil. Energy & Fuels34(7), 7863-7914.
  3. Lu, J. S., Chang, Y., Poon, C. S., & Lee, D. J. (2020). Slow pyrolysis of municipal solid waste (MSW): A review. Bioresource Technology312, 123615.

- The equations are not very clear. The authors should better use MathType or Latex. All equations must be centered.

- Some of the equations do not have all variables explained. For example: eq. (1), (2), (3) and so on.

- All figures must have sources.

- The authors should add a new section Discussion of the results to discuss the results and how they can be interpreted from the perspective of previous studies and of the working hypotheses. The findings and their implications should be discussed in the broadest context possible. Future research directions may also be highlighted.

- The step from eq (5) to eq (6) is not straight forward. The authors should go more in details with that.

- Equation at line 126 should be named eq (13).

As a conclusion of the revision, if all the described suggestions are addressed, the manuscript will reach a better presentation and scientific level, according to the prestigious journal Energies.

Author Response

请参阅附件。

Reviewer 2 Report

The paper was revised according to the journal rules. The topic was focused on  the treatment of special wastes using an alternative method.

Few revisions are required and they are reported below:

  • try to reduce acronyms from the abstract
  • the abstract should contain more quantitative results
  • please add a nomenclature list considering all acronyms and parameters with the unit of measure
  • I suggest to use the same style for all the sections
  • please include also some statistical evaluations - I would include uncertainty analysis
  • describe better table 3 along the manuscript, the role of Tmax and Tf should be clarified
  • few grammar errors should be corrected, check the style, adding also a space between numbers and unit of measure
  • figure 5 should be deeply discussed and compared to the published works
